# Manipulation of Linear Deformable Objects with Non-Negligible Dynamics Grasped at Multiple Locations: A Stability-Guaranteed Approach

*Abstract*— Most existing research on deformable object manipulation assumes the objects are lightweight and exhibit negligible mechanical response, limiting the problem to quasi-static conditions. However, many real-world objects—such as hoses, pipes, and wiring harnesses—have significant dynamics that must be considered during manipulation. This work addresses that gap by proposing a closed-loop control framework that explicitly models object dynamics and treats manipulation as a shape-regulation problem. The system controls the object by adjusting forces and torques at multiple fixed points along its length. The approach is based on three main contributions: a fully dynamic model for linear deformable objects using discrete strain parameterization; an extension of actuation coordinates to SE(3), enabling a structured yet naturally underactuated control framework; and nonlinear feedback control laws with clear conditions for convergence to desired steady-state shapes. Simulation results on representative tasks show clear performance improvements using the proposed model-based method. Experimental validation with real-time closed-loop control and online shape estimation further demonstrates the method's practical feasibility and effectiveness.

## I. INTRODUCTION

The integration of robots into human environments remains bottlenecked by manipulation, and in particular by the inability to robustly handle deformable objects—such as cables, hoses, or groceries—that pervade everyday tasks [1]–[3]. Deformable Linear Objects (DLOs)—including cables, wires, ropes, plants, and sutures—appear across a wide range of applications, from automotive, agricultural, and aerospace manufacturing to household and medical technologies. These objects are slender deformable bodies whose configuration evolves primarily along a single spatial dimension, making them prone to large, continuous deformations during interaction. As a result, even representing the state of a DLO requires, in principle, a theoretically infinite number of degrees of freedom, making its analysis inherently challenging. This difficulty is further exacerbated when object dynamics cannot be neglected—either because stiffness or mass are significant, or because the object undergoes rapid motion. Most existing approaches focus either on model-free learning strategies or on quasi-static, often purely geometric, formulations. In both cases, the object's physical response is not explicitly accounted for [4]–[9]. On the other hand, learning-based strategies have recently become popular for dynamic DLO manipulation, aiming to overcome quasi-static assumptions [10]–[17], but learning accurate global deformation models remains data-inefficient due to strong nonlinear dependence on the DLO configuration. Conse-

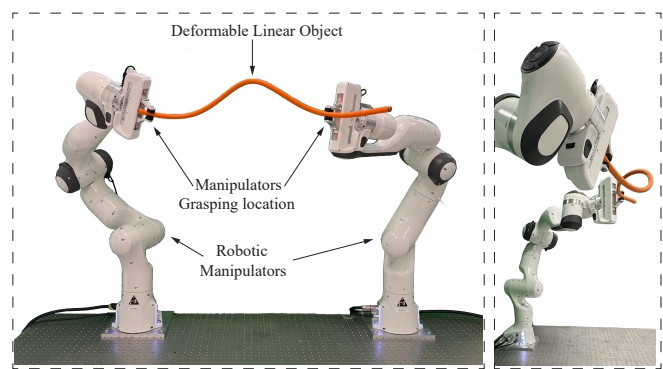

Fig. 1: Manipulation of a deformable linear object (DLO) whose behavior is governed by its physical response and interaction with multiple contact points. Two robotic manipulators apply time-varying wrenches that couple with the object's distributed inertia, elasticity, and gravity, driving the system toward the steady-state configuration shown, characterized by a nontrivial combination of bending, twisting, and shearing. The proposed architecture achieves stable and precise three-dimensional shape control by treating the deformable object itself as the controlled dynamical system and regulating it through model-based feedback.

quently, achieving reliable robotic manipulation of DLOs with guaranteed stability and precision remains an open problem.

This work introduces the first model-based closed-loop control framework for the robotic manipulation of generic DLOs deforming in three dimensions under the action of multiple contact points (see Fig. 1). Manipulation is cast as a regulation problem in which the object itself defines the controlled dynamical system, while the control inputs are wrenches (forces and torques) applied at the contact points. This perspective enables precise shape control with formal guarantees of closed-loop stability. At the core of the approach lies a general dynamic model of DLOs as continuously deformable rods. Building on recent advances in soft robotics [18], [19], the model explicitly captures distributed inertia, elastic and gravitational effects, and treats the object as a floating-base system subject to wrenches applied at multiple, arbitrary locations. A strain-based discretization is then used to obtain accurate yet low-dimensional approximations of the resulting dynamics. On this basis, we develop a general control framework that allow us to design controllers that ensure stability guarantees to the desired equilibrium shape.

## II. DYNAMICS OF A DLO WITH NON-NEGLIGIBLE PHYSICAL RESPONSE

A floating slender soft body of length $L$ can be modeled as a Cosserat rod, represented as a continuous stack of rigid cross-sections parameterized by a curvilinear abscissa $s \in [0, 1]$, where its base is free to move in space. The motion of the base configuration can be described using rigid-body motions, i.e., homogeneous transformation matrices in the Special Euclidean group $SE(3)$. The base pose is represented by the homogeneous transformation matrix $\boldsymbol{g}_{\mathrm{b}} \in SE(3)$, whose generalized coordinates $\boldsymbol{q}_{\mathrm{b}} \in \mathbb{R}^6$ define the orientation and position of the object frame with respect to the global frame. Similarly, the linear and angular velocities of the moving object frame are defined by the body twist $\boldsymbol{\eta}_{\mathrm{b}}$, expressed in body-frame coordinates. Using screw theory, this twist is given by $\boldsymbol{\eta}_{\mathrm{b}} = \boldsymbol{g}_{\mathrm{b}}^{-1} \dot{\boldsymbol{g}}_{\mathrm{b}}$. Next, we define the homogeneous transformation matrix between the coordinate frame attached to each cross-section and the object frame as the directed spatial curve $\boldsymbol{g}_{\mathrm{r}}(\bullet) : s \rightarrow \boldsymbol{g}_{\mathrm{r}}(s) \in SE(3)$ which assigns to each point $s$ along the rod a rigid-body configuration as follows:

$$\boldsymbol{g}_{\mathrm{r}}(s) = \begin{bmatrix} \boldsymbol{R}(s) & \boldsymbol{r}(s) \\ \boldsymbol{0} & 1 \end{bmatrix}, \tag{1}$$

where $\boldsymbol{r}(s) \in \mathbb{R}^3$ is the position of the local frame, while $\boldsymbol{R}(s) \in \mathrm{SO}(3)$ provides the orientation of the local frame. The strain field $\boldsymbol{\xi}(s)$ and the relative velocity twist $\boldsymbol{\eta}_{\mathrm{r}}(s)$ of the body with respect to the object frame are defined by the partial derivatives of Equation (1) with respect to space $(\cdot)'$ and time $\dot{(\cdot)}$, respectively:

$$\boldsymbol{g}_{\mathrm{r}}'(s) = \boldsymbol{g}_{\mathrm{r}} \hat{\boldsymbol{\xi}}(s), \qquad \dot{\boldsymbol{g}}_{\mathrm{r}}(s) = \boldsymbol{g}_{\mathrm{r}}(s) \hat{\boldsymbol{\eta}}_{\mathrm{r}}(s), \tag{2}$$

where $\widehat{(\bullet)}$ denotes the isomorphism between $\mathbb{R}^6$ and the Lie algebra $\mathfrak{se}(3)$, following the screw-theoretic representation of twists. The relation between screw strain and velocity is established through the equality of the mixed partial derivatives in space and time:

$$\boldsymbol{\eta}_{\mathrm{r}}' = \dot{\boldsymbol{\xi}} - \mathrm{ad}_{\boldsymbol{\xi}} \boldsymbol{\eta}_{\mathrm{r}} \tag{3}$$

where $\mathrm{ad}_{(\bullet)}$ is the adjoint operator of $se(3)$. Space integration of equations (2) (first part) and (3) provides the homogeneous transformation and the velocity twist of any point of the object with respect to the spatial frame:

$$\boldsymbol{g}(s) = \boldsymbol{g}_{\mathrm{b}} \cdot (e^{\int_0^s \hat{\boldsymbol{\xi}} d\gamma}); \quad \boldsymbol{\eta}(s) = \boldsymbol{\eta}_{\mathrm{b}} + \mathrm{Ad}_{\boldsymbol{g}^{-1}} \int_0^s \mathrm{Ad}_{\boldsymbol{g}} \dot{\boldsymbol{\xi}} d\gamma \tag{4}$$

where $\mathrm{Ad}_{\boldsymbol{g}}$ is the adjoint representation of $\boldsymbol{g}$. Up to this point, the formulation remains general, and equations (1)–(4) define the PDE system describing the Cosserat rod. We propose employing a "Ritz–Galerkin" method in which the configuration is parameterized by the strain field, discretized using a truncated functional basis of spatially dependent vectors. The components along these basis vectors define a finite set of generalized coordinates, which are governed by a system of Lagrangian ODEs in time.

$$\boldsymbol{\xi}(s, t) \approx \boldsymbol{\xi}_n(s, t) = \boldsymbol{\xi}_0(s) + \sum_{i=1}^n q_{\mathrm{o},i}(t) \, \boldsymbol{\phi}_i(s). \tag{5}$$

where $\boldsymbol{\xi}_n \in \mathbb{R}^6$ is the approximated strain field and encompasses the six deformation modes including bending, twisting, shear, and elongation, $\boldsymbol{\xi}_0(s)$ is the reference strain field, $\boldsymbol{\phi}_i(s) \in \mathbb{R}^6$ are the basis functions, and $\boldsymbol{q}_{\mathrm{o}} = [q_{\mathrm{o},1}, q_{\mathrm{o},2}, ..., q_{\mathrm{o},n}]^T \in \mathbb{R}^{n_{\mathrm{o}}}$ are the generalized coordinates that define the deformation of the object. Equation (5) can be substituted into Equation (3), eventually leading to the definition of the geometric Jacobian of the full system.

$$\boldsymbol{\eta}(s) = \boldsymbol{\eta}_{\mathrm{b}}(\boldsymbol{q}_{\mathrm{b}}) + \mathrm{Ad}_{\boldsymbol{g}}^{-1} \int_0^s \mathrm{Ad}_{\boldsymbol{g}} \boldsymbol{B}_\xi d\gamma \dot{\boldsymbol{q}}_o = \boldsymbol{J}(\boldsymbol{q}, X) \dot{\boldsymbol{q}} \tag{6}$$

where $\boldsymbol{q} = [\boldsymbol{q}_{\mathrm{b}}^T \ \boldsymbol{q}_{\mathrm{o}}^T]^T \in \mathbb{R}^n$ are the generalized coordinates of the floating DLOs ($n = 6 + n_{\mathrm{o}}$) and $\boldsymbol{B}_\xi \in \mathbb{R}^{6 \times n_{\mathrm{o}}}$ is the matrix collecting basis functions column-wise.

Projecting the free dynamics of the Cosserat rod [20] using the geometric Jacobian through D'Alembert's principle derives the generalized dynamics of the system

$$\boldsymbol{M}(\boldsymbol{q})\ddot{\boldsymbol{q}} + (\boldsymbol{C}(\boldsymbol{q}, \dot{\boldsymbol{q}}) + \boldsymbol{D})\dot{\boldsymbol{q}} + \boldsymbol{K}\boldsymbol{q} + \boldsymbol{G}(\boldsymbol{q}) = \boldsymbol{A}(\boldsymbol{q})\boldsymbol{u} , \tag{7}$$

Here, $\boldsymbol{M}(\boldsymbol{q})$ is the generalized mass matrix, $\boldsymbol{C}(\boldsymbol{q}, \dot{\boldsymbol{q}})$ is the Coriolis matrix, $\boldsymbol{D}$ and $\boldsymbol{K}$ are the damping and stiffness matrices, and $\boldsymbol{G}(\boldsymbol{q})$ represents gravity. Assuming Hooke-like linear elastic and viscoelastic constitutive laws makes $\boldsymbol{K}$ and $\boldsymbol{D}$ configuration-independent. The matrix $\boldsymbol{A}(\boldsymbol{q})$ maps the control inputs into generalized forces.

Each control input is a wrench $\boldsymbol{W}_i = [\boldsymbol{\tau}_i^T, \boldsymbol{f}_i^T]^T \in \mathbb{R}^6$, composed of torque and force applied at specific points of the object. The generalized actuation forces are computed by projecting these wrenches through the transpose of the corresponding spatial Jacobians,

$$\boldsymbol{\tau}_{\mathrm{a}} = \sum_{i=1}^{n_{\mathrm{w}}} \boldsymbol{J}_i^{\mathrm{s}}(\boldsymbol{q})^T \boldsymbol{W}_i = \sum_{i=1}^{n_{\mathrm{w}}} (\boldsymbol{J}_{z_i}^{\mathrm{s}})^T (\boldsymbol{q}) \boldsymbol{\tau}_i + (\boldsymbol{J}_{p_i}^{\mathrm{s}}(\boldsymbol{q}))^T \boldsymbol{f}_i, \tag{8}$$

where $\boldsymbol{J}_i^{\mathrm{s}}$ is the spatial geometric Jacobian at the application point. Collecting all wrenches into the input vector $\boldsymbol{u}$ gives a total of $n_a = 6n_w$ actuation inputs, and the actuation matrix is

$$\boldsymbol{A}(\boldsymbol{q}) = \begin{bmatrix} \boldsymbol{J}_1^{\mathrm{s}}(\boldsymbol{q})^T & \boldsymbol{J}_2^{\mathrm{s}}(\boldsymbol{q})^T & \cdots & \boldsymbol{J}_{n_{\mathrm{w}}}^{\mathrm{s}}(\boldsymbol{q})^T \end{bmatrix} \tag{9}$$

## III. COLLOCATED FORM IN LAGRANGIAN SYSTEMS

A Lagrangian system of the form (7) can be conveniently reformulated by separating the dynamic equations of the actuated $\boldsymbol{\theta}_{\mathbf{a}} \in \mathbb{R}^{n_{\mathrm{a}}}$ and unactuated variables $\boldsymbol{\theta}_{\mathbf{u}} \in \mathbb{R}^{n-n_{\mathrm{a}}}$, such that $\boldsymbol{\theta} = [\boldsymbol{\theta}_{\mathbf{a}}^T \ \boldsymbol{\theta}_{\mathbf{u}}^T]^T$. Finding a more suitable representation of the system dynamics can greatly simplify both analysis and control design through an appropriate transformation of the generalized coordinates. In particular, the system can be expressed in a collocated form under an appropriate change of coordinates. This procedure is described in detail in [21]. For our system, we consider a coordinate transformation in which the new actuated coordinates are given by the SE(3)

parameterization of the object poses at the points where the wrenches are applied.

$$\boldsymbol{\theta}(\boldsymbol{q}) = \left[ \boldsymbol{\theta}_{\mathrm{a},1}^T(\boldsymbol{q}) \ \ldots \ \boldsymbol{\theta}_{\mathrm{a},n_{\mathrm{w}}}^T(\boldsymbol{q}) \ \boldsymbol{c}_{\mathrm{u}}^T(\boldsymbol{q}) \right]^T \quad (10)$$

be such that

$$\boldsymbol{\theta}_{\mathrm{a},i}(\boldsymbol{q}) = \begin{bmatrix} \boldsymbol{z}_i(\boldsymbol{q}) \\ \boldsymbol{p}_i(\boldsymbol{q}) \end{bmatrix} \in \mathbb{R}^6, \qquad \boldsymbol{c}_{\mathrm{u}}(\boldsymbol{q}) \in \mathbb{R}^{n-n_{\mathrm{a}}},$$

where $\boldsymbol{z}_i$ is any local parametrization of $SO(3)$ and $\boldsymbol{c}_{\mathrm{u}}$ is any function such that the Jacobian $\partial\boldsymbol{\theta}/\partial\boldsymbol{q}$ is full rank. The full derivation is omitted here for brevity; however, it can be shown that, after performing the coordinate transformation, the decoupled system dynamics can be expressed as:

$$\begin{bmatrix} \boldsymbol{M}_{\mathrm{aa}} & \boldsymbol{M}_{\mathrm{au}} \\ \boldsymbol{M}_{\mathrm{ua}} & \boldsymbol{M}_{\mathrm{uu}} \end{bmatrix} \begin{bmatrix} \ddot{\boldsymbol{\theta}}_{\mathrm{a}} \\ \ddot{\boldsymbol{\theta}}_{\mathrm{u}} \end{bmatrix} + \begin{bmatrix} \boldsymbol{C}_{\mathrm{aa}} & \boldsymbol{C}_{\mathrm{au}} \\ \boldsymbol{C}_{\mathrm{ua}} & \boldsymbol{C}_{\mathrm{uu}} \end{bmatrix} \begin{bmatrix} \dot{\boldsymbol{\theta}}_{\mathrm{a}} \\ \dot{\boldsymbol{\theta}}_{\mathrm{u}} \end{bmatrix} + \begin{bmatrix} \boldsymbol{G}_{\mathrm{a}} \\ \boldsymbol{G}_{\mathrm{u}} \end{bmatrix}$$
$$+ \begin{bmatrix} \boldsymbol{K}_{\mathrm{aa}} & \boldsymbol{K}_{\mathrm{au}} \\ \boldsymbol{K}_{\mathrm{ua}} & \boldsymbol{K}_{\mathrm{uu}} \end{bmatrix} \begin{bmatrix} \boldsymbol{\theta}_{\mathrm{a}} \\ \boldsymbol{\theta}_{\mathrm{u}} \end{bmatrix} + \begin{bmatrix} \boldsymbol{D}_{\mathrm{aa}} & \boldsymbol{D}_{\mathrm{au}} \\ \boldsymbol{D}_{\mathrm{ua}} & \boldsymbol{D}_{\mathrm{uu}} \end{bmatrix} \begin{bmatrix} \dot{\boldsymbol{\theta}}_{\mathrm{a}} \\ \dot{\boldsymbol{\theta}}_{\mathrm{u}} \end{bmatrix} = \begin{bmatrix} \boldsymbol{\Psi}^{-T} \\ \boldsymbol{0} \end{bmatrix} \boldsymbol{u}$$
$$(11)$$

where $\boldsymbol{\Psi}(\boldsymbol{z}) = \mathrm{diag}(\boldsymbol{\Psi}_1(\boldsymbol{z}_1), \ldots, \boldsymbol{\Psi}_{n_{\mathrm{w}}}(\boldsymbol{z}_{n_{\mathrm{w}}})) \in \mathbb{R}^{n_{\mathrm{a}} \times n_{\mathrm{a}}}$ with

$$\boldsymbol{\Psi}_i(\boldsymbol{z}_i) = \begin{bmatrix} \boldsymbol{P}(\boldsymbol{z}_i) & \boldsymbol{0} \\ \boldsymbol{0} & \boldsymbol{I} \end{bmatrix} \in \mathbb{R}^{6\times6}.$$

where $\boldsymbol{P}(\boldsymbol{z}_i)$ can be obtained for any local parametrization $\boldsymbol{z}_i$ of orientation, $\boldsymbol{\omega}_i = \boldsymbol{P}(\boldsymbol{z}_i)\dot{\boldsymbol{z}}_i$,

## IV. Model-based closed-loop control of DLOs

### A. Low-Level Control

The control strategy developed in this work aims to achieve shape regulation, i.e., to drive the system (7)-(9) toward a desired configuration by applying wrenches at the point where the robots grasp the DLO. Different controllers can be implemented in the new dynamics (11), which ensures stability and convergence to the desired shapes of the DLO (see [22] for a proof of stability and convergence conditions in the collocated form). Some of the controllers used in this work are:

$$\boldsymbol{u} = \boldsymbol{\Psi}^T(\boldsymbol{z})\,\boldsymbol{\nu}. \quad (12)$$

$$\boldsymbol{\nu} = \boldsymbol{K}_P(\boldsymbol{\theta}_{\mathrm{a}}^* - \boldsymbol{\theta}_{\mathrm{a}}) - \boldsymbol{K}_D\dot{\boldsymbol{\theta}}_{\mathrm{a}} + \boldsymbol{K}_I\int_0^t \boldsymbol{s}(\boldsymbol{\theta}_{\mathrm{a}}^* - \boldsymbol{\theta}_{\mathrm{a}})d\rho + \boldsymbol{\eta}(\boldsymbol{\theta},\overline{\boldsymbol{\theta}}) \quad (13)$$

Control law (13) guarantees exponential stability, with a fast convergence rate of the actuated coordinates determined by the controller design. The stability proof for the actuated coordinates follows [23], with additional insights introduced in this work.

### B. High-Level Control to achieve a task goal

The manipulation goal is posed as a shape regulation task in task space $\mathbb{R}^m$ defined by an output function $\boldsymbol{h} : \mathbb{R}^n \to \mathbb{R}^m$ that maps the DLO configuration to points or features of interest. The objective is to find a control input $\boldsymbol{u}$ such that

$$\boldsymbol{h}(\boldsymbol{\theta}_{\mathrm{a}},\boldsymbol{\theta}_{\mathrm{u}}) = \boldsymbol{x}^*. \quad (14)$$

where $\boldsymbol{x}^*$ is the desired target (e.g., point position, orientation, relative pose, or full shape). This leads to the constrained optimization problem

$$\min_{(\boldsymbol{\theta},\boldsymbol{u})\in\mathbb{F}} \quad \|\boldsymbol{x}^* - \boldsymbol{h}(\boldsymbol{\theta}_{\mathrm{a}},\boldsymbol{\theta}_{\mathrm{u}})\|_2$$
$$\text{s.t.} \ \boldsymbol{G}_{\mathrm{a}} + \boldsymbol{K}_{\mathrm{aa}}\boldsymbol{\theta}_{\mathrm{a}} + \boldsymbol{K}_{\mathrm{au}}\boldsymbol{\theta}_{\mathrm{u}} = \boldsymbol{u}. \quad (15)$$
$$\boldsymbol{G}_{\mathrm{u}} + \boldsymbol{K}_{\mathrm{ua}}\boldsymbol{\theta}_{\mathrm{a}} + \boldsymbol{K}_{\mathrm{uu}}\boldsymbol{\theta}_{\mathrm{u}} = \boldsymbol{0}.$$
$$\mathbb{F} = \{\underline{a}_i \leq \boldsymbol{\theta}_{\mathrm{a},i} \leq \overline{a}_i, \quad 1 \leq i \leq n\}.$$

The constraints on actuated coordinates help avoid collisions, especially in constrained environments

## V. Simulation results

Numerical simulations were conducted to evaluate the proposed low-level control strategy and validate the effectiveness of the deformable linear object (DLO) model for shape regulation, with particular focus on robustness, stability, convergence, and the role of model-based compensation. The benchmark system consists of a cylindrical beam with predefined geometric and material properties. Two scenarios were considered: a clamped-base case with 6D wrenches applied at the midpoint and endpoint, and a floating-base bimanual manipulation case with 6D wrenches applied at both endpoints. In both cases, the objective was to regulate the positions and orientations of the actuation points toward four target shapes. The clamped configuration employed a two-element discretization to improve deformation accuracy, while the floating-base case was modeled using a single floating-base element. A PD controller augmented with model-based compensation was applied in both scenarios. The results of Fig. 2 demonstrate accurate pose regulation at the actuation points, along with stable and convergent behavior across all target configurations.

## VI. Experimental Evaluation

The proposed methodology computes the grasping wrenches required for robots to dynamically control the shape of a deformable linear object (DLO) and execute manipulation tasks. Experimental identification of a $0.76,\mathrm{m}$ DLO provided the material parameters used in the model: a Young's modulus of $22.07,\mathrm{MPa}$, a density of $3546.6,\mathrm{kg/m}^3$, and an elastic damping coefficient of $2,\mathrm{MPa}\cdot\mathrm{s}$. Based on simulation results, the DLO shape is represented using quadratic-curvature polynomials in both the $y$ and $z$ directions, resulting in a 6-DoF model. The generalized coordinates $\boldsymbol{q}^{\mathrm{e}}$ are estimated for feedback control to compensate for elasticity and gravity effects. A minimization problem is formulated using marker position errors together with the position and orientation errors at the robot grasping points. The availability of an analytic gradient due to our model enables real-time shape estimation, achieving update rates above 150 Hz for the 6-DoF system.

### A. Experimental results

The controller performance is evaluated in a task-oriented experiment where two robots grasp both endpoints of the

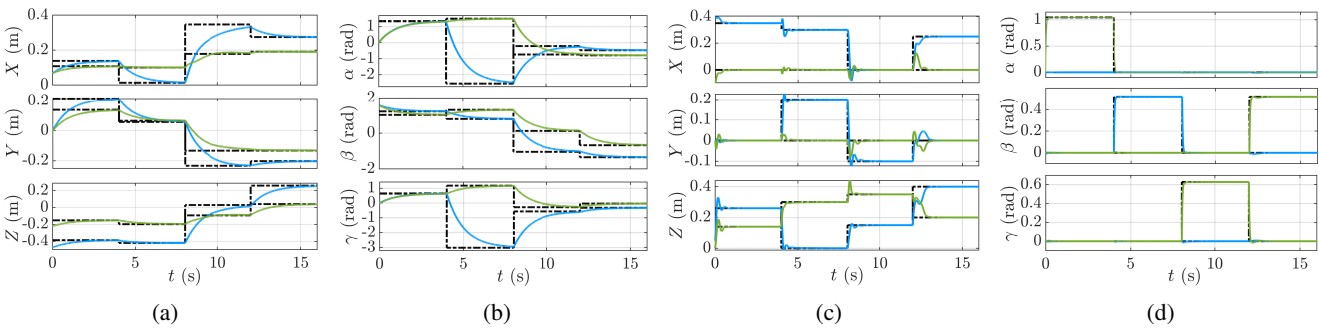

(a)    (b)    (c)    (d)

Fig. 2: Results of closed-loop control of the deformable object: (a)-(b) present the regulation of the actuation coordinates for the first case: the base actuation coordinates are shown in blue, while the endpoint actuation coordinates are shown in green and (c)-(d) shows the regulation of the actuation coordinates for the second case.

deformable object and aim to move its midpoint to predefined target locations. A marker placed at the midpoint is used for visualization, while the control inputs are the 6D wrenches applied at both endpoints ($n_a = 12$). Five target midpoint positions are defined in a constrained scenario. The proposed deformation-aware approach is compared with a model-free baseline that assumes the object is rigid, estimates only its initial shape, and controls the endpoints using a PD controller without considering deformation.

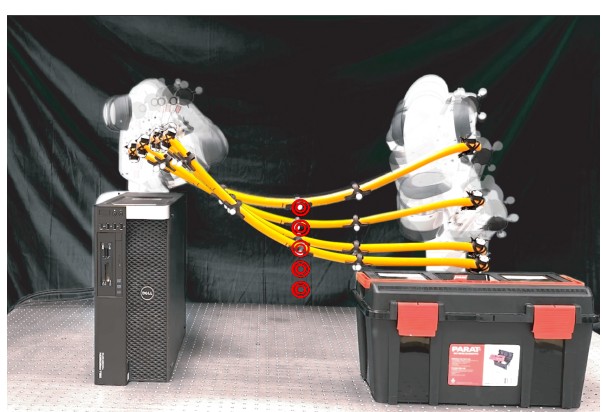

(a)

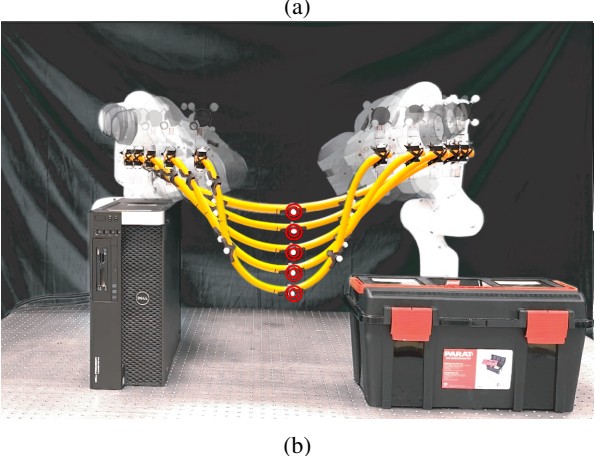

(b)

Fig. 3: Results of the experiments. (a) Frame sequences of the robots' motion using the model-free approach and (b) using the proposed approach. The desired position to be achieved by the midpoint of the DLO is marked with a red circle.

Experimental results (Fig. 3 and Fig. 4), obtained by measuring the midpoint position after executing the optimized configurations, show that the proposed method significantly improves midpoint reachability. By explicitly accounting for deformation, it enables target positions that are not achievable under the rigid-object assumption.

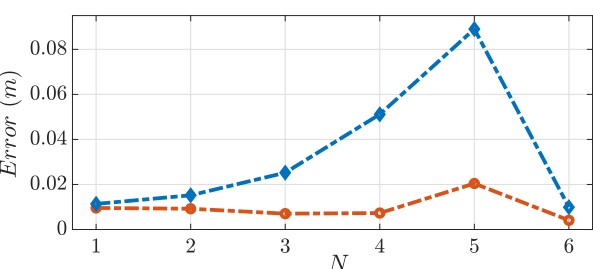

Fig. 4: 3D position components of the object midpoint for both approaches, together with a comparison of the positional errors.

## VII. CONCLUSIONS

This work presents a model-based closed-loop control framework for dynamic manipulation of deformable linear objects using multiple contact points. The object is modeled as a floating-base rod with external forces, capturing elasticity, gravity, and dynamic coupling. Manipulation is formulated as a shape regulation problem with optional Cartesian task objectives. Experiments show high accuracy, real-time performance, and better generalization than model-free methods. Unlike controllers that ignore dynamics and produce steady-state errors, the proposed method achieves zero steady-state error, even for high stiffness and inertia.

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
