# OpenReview forum: "Manipulation of Linear Deformable Objects with Non-Negligible Dynamics Grasped at Multiple Locations: A Stability-Guaranteed Approach"
_IEEE.org/ICRA/2026/Workshop/Manipulation_Robustness — ICRA 2026_

### Official Review · Reviewer_MnLZ · 2026-05-06
**Well-written paper on model-based 3D deformable linear object manipulation**

**Rating:** 7
**Confidence:** 5

**Review:**

The paper presents a model-based closed-loop control framework for dynamic manipulation of 3D deformable linear objects grasped at multiple points, combining a reduced Cosserat-rod dynamics model with collocated nonlinear feedback control to achieve stable, accurate shape regulation with convergence guarantees. The hardware results look promising, and the paper writing is reasonably clear. However, the paper would benefit a lot from defining a lot of variables such as all the partial derivatives in eq(11).The generalizability of the method is also concerning, since the hardware experiments are conducted with a cylindrical beam “with  predefined geometric and material properties”.

---

### Decision · Program_Chairs · 2026-05-21

Accept